# Spatial organisation of catchments - assessment and usage for impartial sub-basin ascertainment and classification

Henning Oppel<sup>1</sup>, Andreas Schumann<sup>1</sup>

<sup>1</sup>Institute of Hydrology, Water Resources Management and Environmental Engineering, Ruhr-University Bochum, Bochum, 5 44801, Germany

Correspondence to: Henning Oppel (henning.oppel@rub.de)

# Abstract.

A hydrological model should represent the hydrological most relevant catchment characteristics. These are heterogeneously distributed within a watershed but often interrelated and subject of a certain spatial organisation. In order to reproduce the

- natural rainfall-runoff response the reduction of variance of catchment properties as well as the incorporation of the spatial organisation of the catchment is desirable. In this study the method of the characteristic structure is introduced to detect and visualize the spatial organisation of catchments, based on stream flow length rearrangement of any catchment feature of interest. Moreover, the method is implemented in an algorithm for automated sub-basin ascertainment, which includes the definition of zones within the newly defined sub-basins. The algorithm is applied on two parameters characterising topography
- and soil of four mid-European watersheds. Results indicate a wide range of applicability for the method and the algorithm. As a limitation of the application for the algorithm the presence of small scale soil enclosures that do not follow the geomorphologic structure of the catchment could be identified. Finally, results of subdivisions based on soil and topography were intersected to gain insight into catchment organisation. Based on this analysis four types of physiographical types could be established.

#### 20 1 Introduction

The identification of landscape units featuring an identical, or at least similar, hydrologic behaviour is a prevailing topic in hydrology. A valid and widely applicable concept is worthwhile for two reasons: for the use in rainfall-runoff-models and the for use in regionalization concepts concerning the prediction in ungauged basins (Sivapalan et al., 2003). Within rainfall-runoff-models these spatial units could define parameter constraints for distributed and semi-distributed models and could,

hence, lower the problem of equifinality (Beven, 2006). On the other hand it would enable a profound method for information transfer from gauged to ungauged basins, or could be used as non-Euclidian distance measure to specify the degree of similarity of two catchments (Skøien et al., 2006).

How these units can be identified is, despite all scientific effort, still unresolved. There are several concepts and findings in the scientific literature. The common foundation for all concepts is the interaction of the physiographic and the hydrological

system within a basin. Winter (2001) expressed that this interaction forms hydrological landscapes, which are defined by landscape, geological framework, climate, surface-, ground- and atmospheric water. The analysis of the hydrologic system is bound to discharge time-series analysis (beside the special case of experimental watersheds) and several catchment classification schemes have been established to identify basins of similar hydrological behaviour (He et al., 2011; Merz and

- Blöschl, 2003; Sawicz et al., 2011; Wagener et al., 2007). The analysis of the physiographic system is more critical than the analysis of the hydrologic system since the driving catchment features responsible for its function are unknown. Studies have shown that catchments with similar soil and topographical values are featuring similar rainfall-runoff dynamics (Müller et al., 2009; Güntner et al., 2004) and therefore most classification schemes try to combine these features (Müller et al., 2009; Dunn and Lilly, 2001; Soulsby et al., 2006). However, these strategies are mostly restricted to a single characteristic. Sivapalan
- (2005) pointed out that the organisation of a catchment is the fundamental influence on the hydrologic system. Patterns of symmetry between soil, topography and the stream network could give insight to underlying mechanisms that induce discharge behaviour. Well known methods to assess the organisation are mainly based on the stream network (Rodríguez-Iturbe and Valdés, 1979; Verdin and Verdin, 1999; Skøien et al., 2006). But these methods neglect information about interdependencies between soil and topography of the catchment. The heights above the nearest drainage concept and classification scheme by
- Nobre et al. (2011) Gharari et al. (2011) and Savenije (2010) takes the geomorphologic allocation of basins cells into account and defines hydrological response units (HRU). This concept does account for soil, topography and their interaction but omits the exact location and possible interactions along the flow path within a basin.

The purpose of this study is to develop methods for impartial und automated definition of sub-basins for modelling purposes and to describe physiographical similarities for these sub-basins. Both methods are supposed to consider the spatial

- organisation of catchments and utilize topographic as well as soil data. At first, we will introduce the method of characteristic structure. For a given catchment characteristic, values are reorganized by their interaction with the flow paths within the basin. With this analysis we are able to assess the succession of the analysed characteristic and the occurrence of regions comprising high or low variance. With the connection to the flow path this analysis brings insight to processes of lateral flow distribution (Grayson and Blöschl, 2001). In contrast to classic methods for spatial pattern evaluation (like Point-by-Point or optimal logical
- alignment methods (Grayson and Blöschl, 2001)) the new method retains information about location and position in the succession of catchment characteristics along the flow path axis. Using this method as a foundation, an algorithm is established to subdivide a basin into several sub-basins in order to reduce the heterogeneity of a given characteristic and reproduce its spatial organisation. Please note that we define sub-basins in the sense of Lindström et al. (1997) as closed hydrological units. Transferred to a hydrological model these sub-basins could be
- used as individual instances and can be linked via flood routing methods. The introduced algorithm incorporates three different techniques for spatial subdivision of a watershed, based on an objective function. The latter identifies regions within a catchment that comprise high variance of a specific catchment characteristic and indicates the need for reduction. With use of three techniques called separation, subdivision at ramifications and zonal

classification this target can be achieved. Each technique is distinctly usable but are used here in succession or in competition to each other in the proposed algorithm.

Results of this algorithm can be used as spatial units for a semi-distributed hydrological model, or via classification for regionalization. In this study we applied the algorithm on four meso-scale catchments in central Europe on characteristics of

5 topography and soil. Results were used to develop four distinct physiographical types. Section 2 will introduce the developed methods and used data. Section 3 presents the results of application. Discussion of obtained results and physiographical types will be presented in Sect. 4. Final conclusions and outlook will be given in Sect. 5.

# 2 Methods and Data

# 2.1 Data

- Four catchments were chosen for application. The basin of the Mulde (Fig. 1, lower right) is located mostly in eastern Germany and with a small part in north Czech. It covers the mid-range mountainous region of the Ore Mountains in the south and the Mulde Loess Loam Hills in the north. With a size of 6170 km<sup>2</sup> it is the largest catchment used in this study. Located in the Bohemain Forest in west Bavaria, the catchment of the river Regen (Fig. 1 upper left) is with a size of 2613 km<sup>2</sup> the smallest river basin used for application. The headwater catchment of the Main (Fig. 1, upper right), including the White and Red Main
- comprises an area of 4224 km<sup>2</sup>. Last selected catchment is an alpine catchment with similar size to the Mulde, the catchment of the Salzach in Salzburg, Austria, comprising an area of 5995 km<sup>2</sup>. All catchment own different geomorphologic structures and river network types. While in the first three catchments, higher mountains are nearly exclusively located at the outer watershed of the catchment, the Salzach catchments contains three big mountains located at the centre of the catchment. The two main tributaries encompass these mountains. While the basin of the Mulde has a nearly continuous increase of slope and heights from north to south, the topography of the remaining catchments is much more heterogeneous.
- heights from north to south, the topography of the remaining catchments is much more heterogeneous. For the proposed methods and algorithm at least a digital elevation model (DEM) is essential. For this study we used a gridded DEM derived from the Shuttle Radar Topography Mission (SRTM) with a regular 100 meter resolution. By means of the D8algorithm the required data like flow directions, flow length and flow accumulation were calculated (Jenson and Domingue, 1988). For the catchment of the Mulde a proved digital river network was available. Stream networks of the remaining basins
- were calculated via flow accumulation algorithms. Because it is envisaged to subdivide the presented catchments by soil characteristics, a gridded soil data map by the German Federal Institute for Geosciences and Natural Resources (BÜK200) and CORINE land coverage data (CLC) (Bossard et al., 2000) were used. Pedo-transfer functions (Sponagel, 2005) combined these information into gridded data about (available) water capacities, hydraulic conductivities and other characteristics. In case of the Salzach basin precast pore volume data for the LARSIM-ME model were used, due to a lack of soil data (Bremicker, 2016).
- Pore volume data is depicted in Fig. 2 for watersheds of the Main, Regen and Salzach. Data for the Mulde basin is shown in Fig. 3. Beside the mentioned topographical structure of the basin we can now see similarities and differences in the pedologic system. The dependency of pore volume and heights is common for all basins, mountainous regions tend to have lower values

than flat lands. A main difference is the arrangement of higher values. While in the Salzach catchment only mid-range values align around the streams and only smaller spots of higher values are present, the catchment of the Mulde has a break between high and low values in the centre of the basin. Moreover, wide soil belts with high pore volume encompass the river valleys in transition area. In comparison to these pattern, soils in the Regen and Main basin seem nearly homogenous with respect to some stripes of higher values near the outlet of the basins.

**2.2 Characteristic structures** 

We defined the spatial organisation of a catchments by analysing the succession of values of selected catchment characteristics along the flow path of a catchment. The flow length within the catchment is evaluated by a pathway oriented search. The lateral flow lengths are divided into flow lengths within the river network to the outlet (called stream flow length (SFL)) and along

- the land surface to the next stream cell (defined as over land flow length (OFL)). Note that the perspective is set upstream. To evaluate the characteristic structures of a catchment characteristic, their values within each grid cell were combined with their SFL and OFL data. Since distances are not continuous, due to the gridded data and depending on grid size, distances were ordered into classes. Therefore width of  $\Delta s$  for SFL- and  $\Delta o$  for OFL classes have to be defined. The basin is thereafter treated as a succession of SFL-classes and OFL-classes.
- All grid-cells of the catchment are merged in their respective class by following routine: all cells feature an SFL value greater or equal than (i-1)· $\Delta$ s and smaller than i· $\Delta$ s are drawn into the cluster i, where i is an integer and raised from one until no cells unassigned are left. For the presented catchments a value for  $\Delta$ s in a range from 1 to 10 km proved to be most adequate. For each SFL-class an average value and standard deviation of the characteristic of interest within the distance class can be calculated:

$$\bar{X}_{sfl} = \frac{1}{n_{Cells;sfl}} \sum_{j=0}^{n_{Cells;sfl}} X_j$$
 (1)

$$\sigma_{sfl} = \sqrt{\frac{1}{n_{Cells;sfl}} - 1} \sum_{j=1}^{n_{Cells;sfl}} \left( \mathbf{X}_j - \overline{\mathbf{X}}_j \right)^2 \tag{2}$$

where sfl indicates the SFL-class, n<sub>Cells,sfl</sub> is the number of cells assigned to this class and X is the characteristic of interest. If the average and the standard deviation of each class is plotted against its average distance to the outlet, the characteristic structure of the catchments is made visible. As an example Fig. 3 shows the available water capacity (AWC) in the Mulde

catchment and its characteristic structure (with  $\Delta s = 1$  km).

An amplification of this method is to take the OFL-classes into account. By their distance to the next stream it is possible to distinguish between regions which are close to stream or more distant and to evaluate the catchment characteristic for these regions separately. In Fig. 4 the characteristic structure of AWC is separated by distance classes of OFL. Instead of the one- $\sigma$ -range, averages of OFL-classes are depicted. From these analysis one might conclude that the variance in the SFL classes (in

Fig.3) is mainly caused by differences of AWC according to the distance from stream cells.

Note that the method can be applied to all gridded data with a metric scale (soil, climate, topography, etc.). Moreover, it is possible to identify shifts of means and variance along the flow path and can allocate their placement within the catchment.

# 2.3 An algorithm for automated sub-basin ascertainment

If we assume that the aim of sub-basin ascertainment is to reduce heterogeneity within the watershed, it is obvious that the 5 method of characteristic structures is a convenient tool for this target. Shifts of variance along the SFL-axis and high variance regions can be interpreted as regions within a catchment that need to be subdivided into sub-basins. From the characteristic structure in Fig. 3 we can derive several cases that need to be handled by an algorithm for sub-basin ascertainment:

- 1. Presence of low variance regions and high variance regions: it is expedient to separate low variance regions from the high variance regions because the former do not require further handling.
- 2. Presence of high variance regions solely: the basin needs to be subdivided in order to decrease the heterogeneity.
  - 3. Presence of low variance regions: no action required.

For separation, two tools and an objective function had to be developed. Task of the objective function is to identify high and low variance regions. For this purpose it utilizes the method of characteristic structures, as described above. A tool for separation of low variance regions needs to allocate the drainage point of target areas, marked by values of the objective

- function below a certain threshold. For the reduction of variance in high variance regions we developed two strategies that are conducted in competition. The first strategy proceeds the subdivision by the identification of confluences, or in this case ramifications, because we are looking upstream. This strategy aims to subdivide a basin by its river network. Second strategy is the definition of zones without the definition of additional sub-basins. Two zone types were introduced: close to stream (indicating wetlands) and far from stream (indicating hillslopes). The definition of zones is bound the river network and the
- OFL data.

All tools are introduced in the following sections. For better understanding their procedures are demonstrated for a synthetic catchment shown in Fig. 5. On the left of Fig. 5 flow directions and Strahler orders are shown, in the middle the SFL data and the SFL-class (here the width of the classes is set to 5) and on the right an arbitrary input variable is shown. After the introduction of the developed tools the sequence of the ACS (Ascertainment by Characteristic Structure) algorithm will be

illustrated.

# 2.3.1 Objective Function

The value of the objective function is calculated from the characteristic structure of the standard deviation  $\sigma$ , shown in Fig. 6. (Additionally the standard deviation is shown for two different values of  $\Delta s$ ). A casual observer will easily spot regions with higher  $\sigma$  that should be considered for ascertainment. To make this feasible for the ACS a threshold value  $\Omega$  has been introduced that states whether a SFL-class is to file as "low" or as "high" variance class. This is derived from the data by:

(3)

$$\Omega = \frac{\sum_{j=0}^{N_{sfl}} \omega_j^{e} \cdot \sigma_j}{\sum_{j=0}^{N_{sfl}} \omega_j}$$

where  $N_{sfl}$  is the number of SFL-classes,  $\sigma$  is the standard deviation within SFL-class i (Eq. (2)), the exponent e is a variable non-linearity factor and  $\omega$  a weighting factor is defined as:

$$\omega_{j} = \frac{\sigma_{j} - \max_{s \not l}(\sigma)}{\min_{s \not l}(\sigma) - \max_{s \not l}(\sigma)}$$
(4)

Coherent SFL-classes above the threshold are considered as high variance regions within the basin that require spatial separation. If regions of SFL-classes below the threshold are found, these regions will be marked as low variance regions. In Fig. 7 two results of an application of the objective function are shown: the first call returns the low variance region in the middle of the catchment and the second call the high variance regions in the lower part.

#### 2.3.2 Separation algorithm for low variance regions

Low variance regions have no need for further subdivisions. Therefore they are separated from the rest of the basin. Since the exact allocation of these regions is known, all cells within can be defined as target area T (hatched in Fig. 7, 1.call, left side). Remaining cells are drawn together as non-target area NT. If one random point of the basin is selected as possible separation point (SP) and its watershed is calculated the set of points belonging to the watershed, or sub-basin, of SP, B<sub>SP</sub> is obtained. The calculated watershed B<sub>SP</sub> covers parts of T and NT and hence a coverage rate can be calculated as the proportion of the cardinalities of the intersections and their respective superset:

$$C_{SP} = \frac{\left|B_{SP} \cap T\right|}{\left|T\right|} - \frac{\left|B_{SP} \cap NT\right|}{\left|NT\right|} \to \max$$
(6)

The objective of a separation C is to find a separation point (SP) whose basin  $B_{SP}$  covers a maximum of T and a minimum of NT. Please note that for regions located at the outlet of the basin or at its upstream boundary only one SP will be defined. Possible SPs are assumed to be allocated at the transit of the main stream from T to NT, or vice versa. An iterative search

# returns the coverage values C and the highest value is selected as SP, defining a new sub-basin. In the upper part of Fig. 7 a concluded separation, as well as the rejected SPs of the iteration (hollow points) are shown.

# 2.3.3 Subdivision at confluences/ramifications

Because the chosen perspective is upstream, the ACS is searching for ramifications of the river network, where a main river splits into two tributaries. To identify ramifications the characteristic structure of flow accumulation (FAcc) is examined. Since

the FAcc indicates the contributing drainage area to each stream cell, discontinuities in the characteristic structure, which is the succession of the FAcc values, will reveal confluences of streams.

Figure 7 shows an example for the Mulde. Beginning at the Outlet (zero on the x-axis) two features are visible: a slowly decreasing line of high FAcc values, representing the main stream at the outlet, and a noise-like smaller range of FAcc values close to the abscise, caused by the smaller tributary streams and contributing areas. To identify major tributaries this noise has to be removed. We assume that in the first distance class the disparity between main rivers and contributing hillslopes is most distinct. Within the first SFL-class a k-Means cluster analysis is carried out to divide high and low FAcc values. Threshold value  $\tau_s$  is determined as:

$$\tau_{s} = \min_{c} \left[ \max_{c1} \left[ FAcc \right]; \max_{c2} \left[ FAcc \right] \right] \cdot \left( 1 - \frac{i}{10} \right)$$
(5)

- where c indicates the clusters and i is the reduction order and by default 0. The algorithm will start with the default value for i and searches for the first ramification in upstream direction. If no ramification is found, order i is increased by 1. The maximum order is set to 10. Please note that the higher i is set, the lower the threshold gets and more FAcc-values remain for analysis. The routine identifies the coordinates of the ramification inducing the drop in FAcc values (see Fig. 8, for example at SFL ≈ 80 km). Two new drainage points are set at the ramification, defining the new sub-basins. The subdivision of a synthetic
- catchment is shown in Fig. 7 (lower middle).

#### 2.3.4 Zonal classification

Another way to subdivide a high variance region without setting new separation points is to split the cells within the region, i.e. the catchment into zones. This is comparable to the concept of HRUs. But, in this case we take the allocation of each cell and its position along the SFL-axis into account. Therefore, only two possible types were implemented: "close to stream" and

20 "far from stream" cells. For some catchment characteristics these zone might comply with natural wetlands and hillslopes, but is not compulsory for all possible input data.

To identify close to stream cells, firstly the stream has to be selected. Since, we try to emulate natural patterns of the input data not all present stream cells might be relevant. We assume that we need to separate data fields around higher order streams from data fields located at lower order streams. Therefore, streams are selected by their Strahler order. Cells draining into the

selected stream cells, with an OFL value equal or below a threshold  $\Delta o$ , are defined as "close to stream" cells. Remaining cells are marked as "far from stream".

Because we do not know the dominant spatial pattern of the input data the search for an optimal extent of the "close to stream" zone is done iteratively. The iteration employs two variables: the reduction of the Strahler order  $s_R$  from the maximum occurring Strahler order  $M_S$  and the width of the zones  $\Delta o$ , expressed as multiple of cell width. Parameter ranges are [0;  $M_S$ -

30 1] for  $s_R$  and [0, 5] for  $\Delta o$ , respectively. Cells draining into streams cells with Strahler order equal or lower than  $M_S$ - $s_R$  and an OFL value equal or lower than  $\Delta o \cdot \Delta x$  are marked as "close to stream". In Fig. 9 a sequence of the iteration is shown for the

entire synthetic catchment. After each iteration, the standard deviation  $\sigma$  (Eq. (2)) is calculated for each zone and subsequently averaged. The parameter combination ( $s_R$ ,  $\Delta o$ ) with the lowest averaged standard deviation is chosen for final classification. In Figure 6, an example for a chosen zonal classification for the synthetic catchment is shown. The ACS will define results of the technique leading to the highest reduction of the standard deviation  $\sigma$  as superior and omits the inferior result.

Please note that the algorithm has the possibility to neglect the usage of zonal classification. If the calculated averaged  $\sigma$  of 5 the zones is equal to or higher than  $\sigma$  of the unseparated data, the whole basin will be marked as zone type "none".

#### 2.3.5 Sequence of the ACS-algorithm

Now that we have introduced all necessary tools we have to compile them into an algorithm that will ascertain sub-basins automatically. At first the targets and restrictions for the ascertainment have be defined: The main target was set to minimise

- $\sigma$  of the considered feature which is expressed in the objective function. We posit that shifts and breaks in the average of the characteristics along the SFL-axis can be compensated within the (rainfall-runoff) model structure using the obtained catchment subdivision. Furthermore the capability to make use of HRUs or zones within each sub-basin is presumed, although the latter presumption is not mandatory. An additional assumption is that the consideration of major streams within a model structure is worthwhile, independent from variance compulsion. In the completed algorithm the separation of major streams is
- made an optional choice. Major streams, or major stream ramifications are identified likewise to any other ramification (Sect. 2.3.3), but additionally the FAcc value of the tributary stream has to be higher than the threshold value  $\tau_R$ . This parameter is calculated as percentage of the maximum FAcc value in the basin (e.g. 5%) once at the initialisation of the algorithm. Figure 10 shows the implemented sequence of the algorithm as a flow chart, as a result of the following considerations:

  - 1. If a major ramification is present, the subdivision at ramification is activated without calling the objective function.
  - 2. The presence of low variance regions requires their separation from the remaining regions and is succeeded preliminary to subdivision.
  - 3. In case of present high variance regions solely, results of the subdivision technique yielding a lower averaged value of  $\sigma$  will be saved, other results will be discarded.

20

- 4. In order to obtain a consistent results, zonal classification is called additionally in case only low variance regions are present. Thereby, independent from the need of  $\sigma$  reduction, all ascertained sub-basins will comprise a zonal
- classification.

Beginning at the outlet of the entire catchment ACS will check at first if a major ramification is present in the catchment (if activated). This being the case, the subdivision at ramification is activated (Sect. 2.3.3) and two new sub-basins are obtained and are stored. Now the basin between the initial outlet point and the newly defined separation points will be analysed again.

The objective function (Sect. 2.3.1) is called and based on its answer decided if a partition is required. If high variance regions 30 are present in the basin actions of sub-basin ascertainment are necessary. Conditional to the presence of low variance regions separation tool (Sect. 2.2.2) or separation tools are called. Otherwise the supplementary zones are calculated (Sect. 2.3.4) and the next basin is analysed.

# **3 Results**

Sub-basin ascertainment has been carried out on two characteristics: pore volume (total pore volume for Mulde, Regen and Main, AWC for Salzach (due to data availability, compare Fig. 2 & Fig. 3)) and surface slope. Please note that that both applications and, hence, their results are disjoint.

- In order to evaluate the success of the applications we have to compare  $\sigma$  before separation (U) and  $\sigma$  after separation (S) for the entire catchment. In case of  $\sigma(U)$  Eq. (2) can be applied without modifications. For  $\sigma(S)$  we have to account for the succeeded sub-division of the catchment. If within a SFL-distance class (of the entire catchment) neighbouring sub-basins are present,  $\sigma$  is calculated individually for each sub-basin and afterwards  $\sigma(S)$  for the entire SFL-class is calculated as area weighted average of these  $\sigma$  values.
- Sub-basins and zones for all basins and applications are depicted Fig. 11. The characteristic structure of  $\sigma(U)$  (blue lines) and  $\sigma(S)$  (red lines) as well as threshold  $\Omega$  (according to Eq.(3)) for different weightings e = 0, e = 0.5 and e = 1 are shown. Outlets of the defined sub-basins are classified by their derivation. Drainage points at major ramifications are depicted as black triangles, while blue dots indicate points for separation of low variance regions and subdivisions at ramifications with order 1. These points are merged as category 1 points. Category 2 points indicate points of subdivision of higher order and hence
- indicate the separation of smaller contributing streams.

Although the success and differences between the outcomes of the algorithm are visible, we are not able to quantify these observations. To make our results commensurable two numerical measures of success were developed. Both measures are based on the objective function that implies a minimisation of  $\sigma$ . This can be expressed as the subtraction of  $\sigma(U)$  and  $\sigma(S)$ . To account for different expected values of the input data (in this case for slope than for storage) the reduction of  $\sigma$  is

20 normalized by the used threshold  $\Omega$ . The first measure is the total reduction  $\alpha_1$ :

$$\alpha_1 = \frac{\sum_{j=0}^{n_{sf}} \sigma_j(U) - \sigma_j(S)}{\Omega}$$
(7)

The second measure was defined as the reduction of  $\sigma$  below the threshold  $\Omega$ . This indicator tells if the target has been achieved just scantly or more solid. Only distance-classes that have reached  $\Omega$  are taken into account for this measure. The set of distance-classes fulfilling this requirement can be written as:

$$25 \quad z = [j_{sfl;1}, \dots j_{sfl;n} \mid \sigma_j \le \Omega]$$

$$\tag{8}$$

Then  $\alpha_2$  can then be written as the reduction of  $\sigma$  in the set of distance-classes z:

$$\alpha_2 = \frac{\sum \sigma_z(U) - \sigma_z(S)}{\Omega}$$
(9)

A summary of the results is given in Table 1. From this table and Fig. 11 we can see that some applications were more successful than others, e.g. application on slope data in the Regen catchment (nearly all classes below threshold, highest  $\alpha_2$ )

compared to slope for the Mulde (nearly no reduction). The density of close to stream zones is comparable for all basins, it ranges from 18 - 25 % for pore volume and 22 - 30 % for slope.

However, the application on pore volume in the Mulde catchment resulted in the highest success rates  $\alpha_1$  and  $\alpha_2$  which is only overcome by the success in the Main catchment. Here we obtained only a small total reduction, but the reduction below  $\Omega$  is superior to all other applications. The lowest pore volume success can be seen in the Regen catchment. Another striking result

5 superior to all other applications. The lowest pore volume success can be seen in the Regen catchment. Another striking result is the outcome in the Salzach catchment. For pore volume, the highest total reduction is achieved, but only little reduction below  $\Omega$ . On the other side application on slope led to high success rated (both measures). These results do not directly indicate that the outcome of the algorithm is bound a specific parameter, hence good and bad results are obtained for both features and different catchments.

# 10 4 Discussion

### 4.1 Limiting factors for the algorithm

Lowest success rates are achieved in the application on slope in the Mulde and pore volume in the Regen catchment. Additionally  $\alpha_1$  of pore volume application in the Salzach catchment is high, but the reduction of  $\sigma$  below the benchmark  $\Omega$  is low. Because the low success rates are not limited to one catchment, or to one specific feature it cannot be conducted that a

15 single catchment structure or a single feature is responsible for inferior outcome. Possible alternative explanatory factors are the geomorphologic structure and/or the specific values of the considered features. To gain more insight on this topic a resampling experiment has been performed. Aiming to examine structural identical

catchments with a different range of featured values. Accounted by their similar size, the Mulde and the Salzach catchment were chosen for resampling. Each cell of the DEM and AWC data has been ranked and cell values were exchanged according

to their rank. Shifting the alpine structure of the Salzach to soil and heights of the Ore Mountains, and the middle mountainous structure of the Mulde to an alpine soil and topography. Resampled AWC values are shown Fig.12. Stream network, SFL- and slope values were recalculated for the new DEM. Afterwards the resampled basins were used for sub-basin ascertainment. Results are shown in Fig. 13 and the success rates are given in Table 2.

One of the least successful separations has been the subdivision of the original Salzach catchment by pore volume. After

- resampling, the same geomorphologic structure with the AWC values of the Mulde basin reaches much higher, in case of  $\alpha_2$  the highest, success rates. On the other side, separation based on resampled slope (derived from resampled DEM) improved for both catchments. This results strengthens the thesis that geomorphologic structure and the individual values are the key factors. If the original DEM values (Fig 1) and the original and resampled AWC values (Fig. 2, 3, 12) are compared, the argumentation gets clearer: In case of the pore volume, small spots comprising much higher values than their surrounding area
- in both catchments are visible. Like enclosures in material sciences, these spots can be considered as soil enclosures within the catchment that do not follow the geomorphologic co-evolutional structure (Blöschl et al., 2013) of the basin.