# Peer review of "Spatial organisation of catchments - assessment and usage for impartial sub-basin ascertainment and classification"

_Hydrology and Earth System Sciences, 2016_

## Referee Comment (RC1) · S. Gharari (Referee) · 1 Oct 2016

I skimmed the manuscript. I found the presented idea very interesting however I did not get couple of main points. I would like to use the opportunity of open discussion to hear the authors' clarification and point of view which can definitely help me in better judgment of the manuscript.

1- I did not understand what the objectives of the study are. I fully understand that the authors try to construct a framework in which the catchment won't be delineated more than necessary or will be delineated as needed. But I did not get what that necessity is dependent on. I believe it is directly related to the choice of the modeling approach or the purpose of delineation in the first place.

[Figure]

2- I missed how the authors validated what they had presented. Is there any evaluation (validation) data which backs up what the authors have suggested and presented in the figures and maps?

3- I am wondering what is "available water capacity AWC" and how it was calculated?

I do not need long answers; few lines of convincing arguments would be sufficient.

Thank you in advance.

With regards

Shervan Gharari

---

## Referee Comment (RC2) · S. Gharari (Referee) · 5 Oct 2016

I would like to thank the authors for their prompt reply.

Regarding the first point made by the authors; I don't agree that the classification can be meaningful without a context. Let's give a simple example. Imagine a case in which we need to group people of a society into bins. From an economical point of view the people in a society can be classified into low, medium and high income groups. If the context is health, then for example weight and height can be used for classification. To have more classes depending on the need we can take into account the blood group of every individual as well. For more sophisticated modeling or planning we may need all the information regarding income, weight, height, education and blood type of every

individual in the society.

The point I want to make here is that classification needs a context a priority. I give a similar example on my own work. In Gharari et al, 2011 we classified a Luxembourgish catchment into three landscape units. However when we started modeling, the delineation to two or three landscape units were not so much different in the context of constrained or calibrated TOPO-FLEX model (Gharari et al, 2014). In fact this was not the classification which told us what the best number of HRUs is, this was the context, in this example the modeling framework, which indicated the needed level of complexity. Going back to my simplified example from the previous paragraph, classification of people into any groups (small or large) doesn't tell me much about the need of that classification and complexity of related modeling effort.

If the authors want to show that their approach is truly showing the optimal complexity in context of rainfall runoff modeling while representing heterogeneity, they should show this using a modeling context or framework such as SWAT or any other available modeling strategy.

Regarding the second point; maybe I missed the point the authors trying to make, but if something cannot be validated it means it cannot be reproducible as well. I am just puzzled how does this strategy work for in a new case. I believe clarification in the text is needed.

Regarding the third point; now it became clear, but I also encourage the authors to state the fact that they comparing the result of two different models. AWC is in itself a model and might be associated with bias and high uncertainty which apparently are ignored here

With regards

Shervan Gharari

---

## Author Comment (AC1) · 5 Oct 2016

Thank you for your interest and your comment. I hope the following points will help to clarify the problems:

1) The spatial heterogeneity of catchments can be considered in different ways in hydrological modelling: by fully distributed or semi-distributed models. The last mentioned modelling approach benefits from the option to subdivide a catchment into subbasins. These sub-basins should be differentiated by the heterogeneities of the hydrological most relevant catchment characteristics. If this heterogeneity is low, neighbouring basins can be modelled together, but if a basin is very different from its neighbours it should be described separately. By analysing this spatial heterogeneity the need for

spatial resolution in modelling can be judged. Within these sub-basins the patterns of hydrological relevant characteristics are interrelated and determined to a certain degree by the landscape. Instead to use combinatorics, these patterns can be used to specify the hydrological spatial units according to flow distances which are used here as order feature. One objective of this study was to automate and objectify the process of sub-basin ascertainment. The second objective was to base the sub-basin ascertainment on the spatial organisation of a watershed.

2) In our opinion there is no impartial method/data for validation, since all subdivisions are either subjective (e .g. manual subdivision) or subject to other objective functions (e.g. based on Stahler order). There are only two aspects that can be proven: applicability to different data (as performed for different data types and basins) and performance (comparison of variance in unseparated and separated basin).

3) The term "available water capacity" follows the nomenclature of the European Soil Database for the maximum amount of water that can be stored in soil and is available for plant growth (Ballabio, et al. 2016). Calculation is based on empirical pedotransferfunctions for soils in Germany (Sponagel 2005), comparable to Wösten, et al. (2001)

Literature: Ballabio C., Panagos P., Montanarella L. Mapping topsoil physical properties at European scale using the LUCAS database (2016) Geoderma, 261 , pp. 110-123. Sponagel, H. (Ed.): Bodenkundliche Kartieranleitung: Mit 103 Tabellen und 31 Listen, 5., verb. und erw. Aufl, 10 Schweizerbart, Stuttgart, 438 pp., 200 (in German) Wösten, J.H.M., Pachepsky, Y.A., Rawls, W.J., 2001. Pedotransfer functions: bridging the gap between available basic soil data and missing soil hydraulic characteristics. J. Hydrol. 251, 123–150.

---

## Author Comment (AC2) · 6 Oct 2016

Regarding your first point, the missing context, we have to point out that we omitted a direct context on purpose. The manuscript is focused on the flow path orientated assessment and the sub-basin ascertainment algorithm. Or in general on the methodology. Our case studies are supposed to test its applicability on different catchment characteristics and geomorphologic structures. The context which the methodology is applied to is up the user. For our case study we assumed a context (pore volume and slope as the driving forces for a semi-distributed model or catchment classification) and applied the algorithm. The obtained results are not believed to be the optimal complexity for a model, but they are the best results in terms of heterogeneity of these selected

characteristics.

As mentioned in the Conclusions of the manuscript, we are aware that we have to demonstrate the benefits of the presented methodology for different purposes. This includes a modelling and a regionalisation experiment. We are currently working on these tasks and results will be published soon. However we do not intend to apply this algorithm for a subdivision of river basins with an existing model (e.g. HBV) but to develop a new modelling approach which can make use from the results in an optimal way. In this sense a modelling application would be too early and in our opinion beyond the scope of this paper. Here we focus on the proposed methods.

Regarding the topic of validation our point has not been expressed clearly. The proposed methods are independent from the case study. Nevertheless, the used threshold value ($\tau$R) and parameter (e) are, indeed, derived from the Mulde catchment. We added a line in the manuscript (data section) that the basin of the Mulde is our development basin and the three remaining catchments are used for unbiased application to demonstrate the generality of this approach. Our previous statement accounted for a missing observation for the validation of the results obtained from the algorithm.

Uncertainties arising from the data are indeed neglected in this study. A note about the uncertainties of input data, concerning all spatial information (soil data and topography data) is added to the data section.

---

## Referee Comment (RC3) · Anonymous Referee #2 · 20 Oct 2016

The Authors propose a network-based classification analysis for dividing hydrologic catchments into more homogeneous sub-catchments.

I agree with S. Gharari on the fact that the idea seems interesting but the paper would be much more convincing if hydrological "validation" would have been performed, i.e., showing that using a semidistributed model accounting for the separations obtained here outperforms other separation schemes. I understand from the Authors replies that this will be not investigated here but rather in following papers. Since this paper essentially is the explanation of an algorithm and the demonstration of its applicability rather than its usefulness, I would suggest to submit the paper as a technical note rather than having it as a research article. This is the reason why I recommend rejec-

tion.

Also, the paper, in its present form, is quite hard to read. In the explanation of the algorithm, the flow of thoughts is, for me, quite confusing. For instance, the "method of characteristic structures" is never really defined (e.g., at page 5, line 5, the Authors state that the method is an obvious choice for their target but it is unclear of what method they are talking about). Also, the amount of details given in some parts of methodology section 2 is sometimes overwhelming and it is easy for the reader to miss the big picture. I would suggest to have a bullet point section (or a diagram) just at the beginning of Section 2, e.g., after the Data subsection, that explains what the following sections will present in detail.

Detailed comments:

Page 2, line 5: please define what the physiographic system is. And what does "more critical" mean?

Page 5, line 5: the "method of characteristic structures" has not been defined yet, therefore I do not see how obvious it is to use it.

Page 5, line 13: again, where is the "method of characteristic structures" defined?

Page 7, line 3: do you mean Figure 8? (see also following references to figures, which seem in many cases wrong)

Page 9, lines 5-10: I would suggest to add here an equation that explains how the sigma after separation is calculated, instead of using just words

Page 10, line 12: do you mean "...has been performed aiming..."?

Page 10, lines 17-22: I cannot understand exactly how the resampling has been done. I would suggest to explain it better.

Page 10, line 27: this sentence is confusing? What do you mean by "geomorphologic structure and the individual values"?

Page 11, line 21: do you mean "spatial extent"? If so, correct also the following occurrences of the wording "spatial extend"

Page 12, bottom and page 13 top: the final separation in types A, B, C and D is actually very interesting but, if I understand it well, seems a rather subjective choice. Since one of the reasons why the Authors propose this method is because of its repeatibility/objectivity, the fact that the "objective" algorithm is used here to help the user to take a final subjective choice should be discussed.

---

## Author Comment (AC3) · 22 Nov 2016

Reviewers' major comment 1: Missing validation of the proposed method by means of a hydrological model, therefore Technical note rather than research article.

- Response -

As already discussed with S. Gahari, we acknowledge the need to demonstrate the opportunities of the proposed method for hydrologic applications. These are not exclusive hydrological modelling but also regionalisation issues. By the application of the proposed methodology in the framework of one or more hydrological models, we would only gain information about the interplay of the GIS-based analyses with the particularly used model structure. The opportunities to parameterise the chosen model and other benefits from our method are closely related with its model structure.

If we interpret your request on a more practical application as a demand for a common way of comparison, with a result that is evident for readers, we propose the following additional analysis: we subdivide the used catchments by the existing gauging network. Obtained subdivisions can be compared to the outcomes of our method. Additionally we will apply a common approach for further zonal subdivision (based on land cover as proposed by (Lindström et al., 1997)). With this analysis we could compare our results on the one hand with a benchmark (nevertheless we are aware that hydrometric networks fulfil also other tasks and are not designed to capture different hydrological characteristics of landscapes only) and on the other hand we establish a further field of application. Results from the algorithm can be used to examine existing gauging networks or as suggestion for spatial resolution for models/ hydrometric networks in ungauged basins. To demonstrate this, we added a new section (4.3) for the paper. New section is placed after line 10 on page 12 in the original manuscript. Moreover results from the new section were incorporated into the conclusions.

Reviewers' major comment 2 and detailed comments 2  3: Explanation of the algorithm is detailed and hard to read.

- Response -

The methods sections intended to describe the methods and algorithm in a way other researcher were able to reproduce its functionalities. The high degree of detailing goes to the expense of intelligibility. To enhance intelligibility the method section of the revised version has been split into a description of objectives and functionality in Section 2 and a more detailed explanation of procedures in an appendix. Additionally, we changed the performance measures $\alpha 1$ and $\alpha 2$ (Eq. 7  9 in the original manuscript) to ratios of variance to make results more palpable. Discussion sections have been adjusted to new evaluation numbers and revised for better intelligibility

Since these points required major revisions we uploaded a new version of the manuscript to the supplement of this commentary.

Detailed comments (Reviewers' Comments (C), Authors Response (A))

C1: please define what the physiographic system is. And what does "more critical" mean? A1: Following passage has been inserted on page 2, line 5: "Contrary to analysis of the hydrologic system where the crucial variable (discharge) is known, the variable(s), yet the number of variables causing the characteristic hydrological behaviour are mostly unknown. This makes the analysis of the physiographic system (the interplay of topography, soil land cover, etc.) more complex than the analysis of the hydrologic system"

C2 C3: Already addressed in response to major comment 2.

C4: Page 7, line 3: Wrong references. A4: References have been updated in course of revision.

C5: "Page 9, lines 5-10: I would suggest to add here an equation that explains how the sigma after separation is calculated, instead of using just words" A5: New section including equation has been added to the new methods section.

C6: "Page 10, line 12: do you mean "...has been performed aiming..."?" A6: We assume denoted reference actually aims to line 17. Passage has been revised.

C7: "I cannot understand exactly how the resampling has been done. I would suggest to explain it better" A7: Explanation of resampling experiment has been extended as follows: "The basic concept is to examine structural identical catchments with a different range of featured values. First step is to examine the spatial organisation of pore volume in the Mulde as it is. Then we change the specific values of pore volume and repeat the analysis. If performance values are similar, the assumption about dependency of performance on the range of values has to be rejected. In order to change the characteristic values in a reasonable way, we did not change the values randomly but

exchanged data of two natural catchments. Accounted by their similar size, the Mulde and the Salzach catchment were chosen for resampling. The exchange of pore volume between these two catchments has to retain the order and arrangement of the original catchments. Therefore we assigned high pore volume values of the original basins to high pore volume values in the exchange basin and applied this scheme to all values. Then the values were exchanged. By this procedure sequences along the flow path like: first high pore volume then low pore volumes have been restored, but the actual pore volume values has changed. The same exchange of values has been applied to the DEM, as the root of slope values. Figurate, the alpine structure of the Salzach has been shifted to soil and heights of the Ore Mountains, and the middle mountainous structure of the Mulde to an alpine soil and topography. If the performance in these resampled basins is identical to the performance in their origin basin (e.g. resampled AWC Mulde and original AWC Salzach) the assumption about dependency of performance on geomorphologic structure has to be rejected."

C8: "Page 10, line 27: this sentence is confusing? What do you mean by "geomorphologic structure and the individual values"?" A8: Geomorphology addresses the interplay of spatial arrangement of topography, soil and other catchment characteristics. The actual values of the single characteristics for example the amount of pore volume are addressed in this sentence. Expanded explanation is included in the beforehand (A7) cited excerpt.

C9: "Page 11, line 21: do you mean "spatial extent"? If so, correct also the following occurrences of the wording "spatial extend"" A9: Misspellings have been corrected.

C10: Subjective choice for classification in Sec 4.3, should be discussed. A10: Following passage has been added on page 13, after line 17 in the original manuscript: "In summary the automatically ascertained sub-basins and zones have been used to categorize regions of catchments into different physiographical types. These types were designed to represent different surface and soil patterns. For the actual categorisation we used the density of defined zones and used gathered information

(Sec. 4.2) about the link of stream network patterns and zone density to derive a classification scheme. However, the absence of an impartial threshold required the (more or less) subjective choice a threshold value for classification of the sub-basins. Therefore the presented results have to be considered as a prospect to future work and possible applications of the algorithm."

Please also note the supplement to this comment:
http://www.hydrol-earth-syst-sci-discuss.net/hess-2016-486/hess-2016-486-AC3-supplement.pdf

[Figure]

**Supplement:**

**Spatial organisation of catchments - assessment and usage for impartial sub-basin ascertainment and classification**

Henning Oppel[1], Andreas Schumann[1]

[1]Institute of Hydrology, Water Resources Management and Environmental Engineering, Ruhr-University Bochum, Bochum, 44801, Germany

*Correspondence to*: Henning Oppel (henning.oppel@rub.de)

**Abstract.**

A hydrological model should represent the hydrological most relevant catchment characteristics. These are heterogeneously distributed within a watershed but often interrelated and subject of a certain spatial organisation. In order to reproduce the natural rainfall-runoff response the reduction of variance of catchment properties as well as the incorporation of the spatial organisation of the catchment is desirable. In this study the method of the characteristic structure is introduced to detect and visualize the spatial organisation of catchments, based on stream flow length rearrangement of any catchment feature of interest. Moreover, the method is implemented in an algorithm for automated sub-basin ascertainment, which includes the definition of zones within the newly defined sub-basins. The algorithm is applied on two parameters characterising topography and soil of four mid-European watersheds. Results indicate a wide range of applicability for the method and the algorithm. As a limitation of the application for the algorithm the presence of small scale soil enclosures that do not follow the geomorphologic structure of the catchment could be identified. Finally, results of subdivisions based on soil and topography were intersected to gain insight into catchment organisation. Based on this analysis four physiographical types of different sub-basin spatial organisations could be established.

**1 Introduction**

The identification of landscape units featuring an identical, or at least similar, hydrologic behaviour is a prevailing topic in hydrology. A valid and widely applicable concept is essential by two reasons: for the use in rainfall-runoff-models and the use in regionalization concepts concerning the prediction in ungauged basins (Sivapalan et al., 2003). Within rainfall-runoff-models these spatial units could define parameter constraints for distributed and semi-distributed models and could, hence, lower the problem of equifinality (Beven, 2006). On the other hand it would enable a profound method for information transfer from gauged to ungauged basins, or could be used as non-Euclidian distance measure to specify the degree of similarity of two catchments (Skøien et al., 2006).

How these units can be identified is, despite all scientific effort, still unresolved. There are several concepts and findings in the scientific literature. The common foundation for all concepts is the interaction of the physiographic and the hydrological

system within a basin. Winter (2001) expressed that this interaction forms hydrological landscapes, which are defined by landscape, geological framework, climate, surface-, ground- and atmospheric water. The analysis of the hydrologic system is bound to discharge time-series analysis (beside the special case of experimental watersheds) and several catchment classification schemes have been established to identify basins of similar hydrological behaviour (He et al., 2011; Merz and

5 Blöschl, 2003; Sawicz et al., 2011; Wagener et al., 2007). Contrary to analysis of the hydrologic system where the crucial variable (discharge) is known, the variable(s), yet the number of variables causing the characteristic hydrological behaviour are mostly unknown. This makes the analysis of the physiographic system (the interplay of topography, soil land cover, etc.) more complex than the analysis of the hydrologic system. Studies have shown that catchments with similar soil and topographical values are featuring similar rainfall-runoff dynamics (Müller et al., 2009; Güntner et al., 2004) and therefore

10 most classification schemes try to combine these features (Müller et al., 2009; Dunn and Lilly, 2001; Soulsby et al., 2006). However, these strategies are mostly restricted to a single characteristic. Sivapalan (2005) pointed out that the organisation of a catchment is the fundamental influence on the hydrologic system. Patterns of symmetry between soil, topography and the stream network could give insight to underlying mechanisms that induce discharge behaviour. Well known methods to assess the organisation are mainly based on the stream network (Rodríguez-Iturbe and Valdés, 1979; Verdin and Verdin, 1999; Skøien

15 et al., 2006). But these methods neglect information about interdependencies between soil and topography of the catchment. The heights above the nearest drainage concept and classification scheme by Nobre et al. (2011) Gharari et al. (2011) and Savenije (2010) takes the geomorphologic allocation of basins cells into account and defines hydrological response units (HRU). This concept does account for soil, topography and their interaction but omits the exact location and possible interactions along the flow path within a basin.

20 In this study we will present a method for automated sub-basin ascertainment based on the spatial organisation of a watershed. First we will introduce the required basic theories and techniques for such a method. A flow path orientated tool for the assessment of spatial organisation of catchments, called the method of characteristic structures will be introduced in this part of the paper. Afterwards this method will be incorporated into an algorithm to perform a subdivision of a catchment based on its individual spatial organisation. The proposed methods are based on a flow path orientated rearrangement of catchment

25 characteristics. With the connection to the flow path this analysis brings insight to processes of lateral flow distribution (Grayson and Blöschl, 2001). In contrast to classic methods for spatial pattern evaluation (like Point-by-Point or optimal logical alignment methods (Grayson and Blöschl, 2001)) the new method retains information about location and position in the succession of catchment characteristics along the flow path axis.

After the outline of the basic theory in Sec. 2 we address the general applicability of the proposed method. Therefore the

30 algorithm was applied to soil and topography characteristics of four meso-scale mid-European watersheds. Based on this results we will evaluate the performance of the algorithm and examine if boundaries for its application are in evidence. In course of this assessment a numerical experiment was carried out to examine different catchment structures with an identical range of values (topography and soil). Results of the application are presented in Sec. 3 and the discussion can be found in Sec. 4.

Beside its general applicability its value for different hydrological applications is addressed. On the one hand we compare automatically ascertained sub-basins to sub-basins defined by a common approach. On the other hand we give a prospect to the use in regionalisation and catchment classification topics. Based on spatial patterns visible in the outcome of the algorithm we established a threshold-based classification scheme to identify different physiographical types.

5  With this concept of the paper we tend to introduce our methods, proof its applicability and give references to possible hydrological applications. The final conclusion and a summary of the obtained results will be given in Sec. 5.

**2 Methods and Data**

The following sections intend to present data, catchments and methods used in this study. First subsection will describe the data. Afterwards a newly developed method for the assessment of spatial organisation of catchments will be introduced. Finally

10  a brief introduction to the implementation of the beforehand proposed methods into an algorithm for automated sub-basin ascertainment will be given.

**2.1 Data**

Four catchments were chosen for application. The basin of the Mulde (Fig. 1, lower right) is located mostly in eastern Germany and with a small part in north Czech. It covers the mid-range mountainous region of the Ore Mountains in the south and the

15  Mulde Loess Loam Hills in the north. With a size of 6170 km$^2$ it is the largest catchment used in this study. Located in the Bohemain Forest in west Bavaria, the catchment of the river Regen (Fig. 1 upper left) is with a size of 2613 km$^2$ the smallest river basin used for application. The headwater catchment of the Main (Fig. 1, upper right), including the White and Red Main comprises an area of 4224 km$^2$. Last selected catchment is an alpine catchment with similar size to the Mulde, the catchment of the Salzach in Salzburg, Austria, comprising an area of 5995 km$^2$. All catchment own different geomorphologic structures

20  and river network types. While in the first three catchments, higher mountains are nearly exclusively located at the outer watershed of the catchment, the Salzach catchments contains three big mountains located at the centre of the catchment. The two main tributaries encompass these mountains. While the basin of the Mulde has a nearly continuous increase of slope and heights from north to south, the topography of the remaining catchments is much more heterogeneous.

For the proposed methods and algorithm at least a digital elevation model (DEM) is essential. For this study we used a gridded

25  DEM derived from the Shuttle Radar Topography Mission (SRTM) with a regular 100 meter resolution. By means of the D8-algorithm the required data like flow directions, flow length and flow accumulation were calculated (Jenson and Domingue, 1988). For the catchment of the Mulde a proved digital river network was available. Stream networks of the remaining basins were calculated via flow accumulation algorithms. Because it is envisaged to subdivide the presented catchments by soil characteristics, a gridded soil data map by the German Federal Institute for Geosciences and Natural Resources (BÜK200) and

30  CORINE land coverage data (CLC) (Bossard et al., 2000) were used. Pedo-transfer functions (Sponagel, 2005) combined these information into gridded data about (available) water capacities, hydraulic conductivities and other characteristics. In case of

the Salzach basin precast pore volume data for the LARSIM-ME model were used, due to a lack of soil data (Bremicker, 2016). The used soil and topography input data comprises a certain amount of uncertainty, due to the fact that they are derived data. The uncertainty of the input data is neglected in the performed case studies. Pore volume data is depicted in Fig. 2 for watersheds of the Main, Regen and Salzach. Data for the Mulde basin is shown in Fig. 3. Beside the mentioned topographical

5   structure of the basin we can now see similarities and differences in the pedologic system. The dependency of pore volume and elevation is common for all basins, mountainous regions tend to have lower values than flat lands. A main difference is the arrangement of higher values. While in the Salzach catchment only mid-range values align around the streams and only smaller spots of higher values are present, the catchment of the Mulde has a break between high and low values in the middle of the basin. Moreover, wide soil belts with high pore volume encompass the river valleys in transition area. In comparison to

10   these pattern, soils in the Regen and Main basin seem nearly homogenous with respect to some stripes of higher values near the outlet of the basins. Please note that the basin of the Mulde is our development basin. Parameters and thresholds used in the case study are derived from this basin. The remaining basins are used for validation of the proposed method.

**2.2 The method of characteristic structures**

River discharge is the result of interactions between atmospheric inputs and the landscape. This interaction can be interpreted

15   as the way water moves vertically and horizontally through the watershed towards the outlet. Differences between catchments, not arising from different atmospheric inputs, are therefore result of different flow path characteristics. If we consider a specific catchment characteristic, like soil or topography, the succession of these values along the flow path within the catchment can be interpreted as the organisation of the watershed.

Flow paths can be detected by means of a DEM and the D8-algorithm (Jenson and Domingue, 1988) and are quantified as

20   length to the drainage. In this study the flow length are differentiated by their surrounding medium. Lateral flow lengths are divided into flow lengths within the river network to the outlet (called stream flow length (SFL)) and along the land surface to the next stream cell (defined as over land flow length (OFL)). Note that the perspective is set upstream.

In order to characterise the succession of soil, topography or other characteristics we have to assign these values to each point of the flow path. In a catchment where each flow length is unique all occurring values of the catchment characteristic of interest

25   can be used as the description directly. Since this is not the case in natural catchments, where multiple parallel streams and hillslopes are present, all equidistant points within the catchment have to be drawn together. With the use of average values and standard deviation (one-sigma range) of equidistant points the succession of catchment characteristics along the flow path can be assessed. Results of the described procedure will be addressed as a "characteristic structure". As an example Fig. 3 shows the map of available water capacity (AWC) in the Mulde catchment and its characteristic structure.

30   Averages and the one-sigma range of standard deviations are plotted as continuous lines, yet underlying data is point wise. Caused by the non-continuous solution (consequence of gridded data and depending on grid size) of distance data. For compensation distances are ordered into classes. Parameter Δs for SFL- and Δo for OFL length are width-thresholds utilised to define distance classes. The basin is thereafter treated as succession of SFL- and OFL-classes. All grid cells of the catchment

are merged in their respective distance-class (for the presented catchments a value for Δs in a range from 1 to 10 km proved to be most adequate). Within each SFL-class an average value and the standard deviation of the characteristic of interest can be calculated:

$$\bar{X}_{sfl} = \frac{1}{n_{Cells;sfl}} \sum_{j=0}^{n_{Cells;sfl}} X_j \qquad (1)$$

$$\sigma_{sfl} = \sqrt{\frac{1}{n_{Cells;sfl}-1} \sum_{j=1}^{n_{Cells;sfl}} \left(X_j - \bar{X}_j\right)^2} \qquad (2)$$

where sfl indicates the SFL-class, $n_{Cells,sfl}$ is the number of cells assigned to this class and X is the characteristic of interest. If the average and the standard deviation of each class is plotted against its average distance to the outlet, the characteristic structure of the catchments is made visible (as shown in Fig. 3).

An extension of this method is to take the OFL-classes into account additionally. By their distance to the next stream it is possible to distinguish between regions which are close to stream or more distant and to evaluate the catchment characteristic for these regions separately. In Fig. 4 the characteristic structure of AWC is separated by distance classes of OFL (width of Δo = 100m). Instead of the one-σ-range, averages of OFL-classes are depicted. From these analysis one might conclude that the variance in the SFL classes (in Fig.3) is mainly caused by differences of AWC according to the distance from stream cells.

Note that the method can be applied to all gridded data with a metric scale (soil, climate, topography, etc.). Moreover, it is possible to identify shifts of means and variance along the flow path and can allocate their placement within the catchment.

**2.3 An algorithm for automated sub-basin ascertainment**

The example characteristic structure in Fig. 3 exhibits that in some parts of the basin equidistant regions comprise similar AWC-values and, hence, the standard deviation is low. In other regions, especially in the centre of the basin, the equidistant regions are very heterogeneous and comprise high standard deviation.

For most hydrological purposes (modelling, regionalisation, search for hydrological units, etc.) homogenous sub-basins are essentially required. Therefore it seems likely to use the proposed method to identify the need for division of a catchment into sub-basin.

For the extension of the method of characteristic structures into an algorithm for sub-basin ascertainment several functionalities have to be introduced:

1. An Objective function to identify the need and position of necessary subdivisions.
2. A Subdivision tools for the actual partition of the catchment.
3. An evaluation strategy for performed subdivisions.

The following subsections will give a brief introduction to these three functionalities, afterwards the sequence of the final algorithm will be described. For more details about the introduced tools please see Appendix A.

**2.3.1 Objective Function**

The value of the objective function is calculated from the characteristic structure of the standard deviation σ, shown in Fig. 5. A casual observer will easily spot regions with higher σ that should be considered for ascertainment. To make this feasible for the algorithm a threshold value Ω has been introduced that states whether a SFL-class is to categorise as "low" or as "high" variance class. Threshold Ω is derived from the data by:

$$\Omega = \frac{\sum_{j=0}^{N_{sfl}} \omega_j^{\,e} \cdot \sigma_j}{\sum_{j=0}^{N_{sfl}} \omega_j} \tag{3}$$

where $N_{sfl}$ is the number of SFL-classes, σ is the standard deviation within SFL-class j (Eq. (2)), the exponent e is a variable non-linearity factor and ω a weighting factor is defined as:

$$\omega_j = \frac{\sigma_j - \max_{sfl}(\sigma)}{\min_{sfl}(\sigma) - \max_{sfl}(\sigma)} \tag{4}$$

As index $N_{SFL}$ in Eq. (3) indicates threshold Ω is calculated for the entire catchment and does not necessarily represent the true value of σ for "homogeneous" sub-basins/SFL-classes. It only delimits between present high and low variance. Dependent on the purpose it might be advisable to recursively apply the proposed algorithm for further reduction of Ω and hence the remaining variance. Note that each application will lead to a higher number of subdivisions.

Coherent SFL-classes above the threshold are considered as high variance *regions* within the basin that require spatial separation. If regions of SFL-classes below the threshold are found, these regions will be marked as low variance regions.

**2.3.2 Subdivision tools**

Figure 6 introduces a synthetic catchment, its stream network, distance classes and a sample catchment characteristic which can be used as input data (for the algorithm under development). The application of the objective function has three possible outcomes:

1. All SFL-classes are below threshold Ω,
2. Parts of the SFL-classes are below Ω,
3. No SFL-classes are below Ω.

First case indicates that no subdivision is required. Consequently no tool has to be applied. Second case indicates that a coherent part of the flow path comprise homogenous characteristics. This case is depicted in the upper left part of Fig. 7. Hatched cells indicate the low variance region. Since this region is homogenous it will not require any further handling and should be clipped from the rest of the basin.

The first developed tool is called *Separation*, since its task is to separate low variance regions from high variance regions. To comply with its task the tool searches the ideal drainage points to capture a maximum of the low variance regions. The search for these separation points is made iteratively close to the transition from low variance SFL- to high variance SFL-classes. In the upper middle of Fig. 7 the result of *Separation* is shown for the introduced synthetic catchment. Black dots indicate the

5    identified drainage points, hollow points indicate rejected points. After applying the *Separation* tool, three sub-basins were defined. One containing the low variance regions that do not require any further treatment and the other two sub-basins comprising the remaining parts of the catchment.

These heterogeneous sub-basins will most likely be subject to the third part of the structuring procedure. In this case all SFL-classes are coherently above the threshold. High variance of catchment characteristics within a watershed can arise from

10   different sources. On the one hand, parallel streams or more specifically neighbouring valleys with different vegetation, slope, etc. will cause high variance in the characteristic structure of the entire basin. On the other hand wetlands close to stream often expose different characteristics as the contributing hillslopes. These patterns are a result of the co-evolutional formation of catchments (Blöschl et al., 2013; Sivapalan, 2005). These stream network orientated patterns for pore volume are visible e. g. in Fig. 2 and Fig. 3.

15   To account for these two origins of heterogeneity, two tools were developed. The first tool is *Subdivision at ramification* which defines new sub-basins at confluences, or ramifications points because our perspective is upstream, of higher order streams. The second tool is a *zonal classification* scheme. Cells within a basin are separated by their distance to their draining stream cell and the respective Strahler order. Distance and order are defined iteratively and obtained zones are called "close to stream -" and "far from stream zones" dependent on their distance to the stream. In the lower part of Fig. 7 both tools are applied to

20   the exemplary synthetic catchment. Hatched cells in the lower left depiction are high variance regions that require partition. Results of the *Subdivision at ramification* tool gives two additional drainage points and basins, results of the *zonal classification* gives no new drainage points but rather two zones belonging to the beforehand defined drainage points.

**2.3.3 Evaluation**

If one beforehand introduced tools has been applied to lower the standard deviation σ in the watershed, it has to be evaluated

25   if this target has been achieved. Therefore we have to reinterpret the objective function. The objective function has been defined to minimise the deviation of a catchment characteristic σ within each distance class. A tool is activated if σ is above threshold Ω. After its application two, or more, parallel basins are present in the same distance class. Note that the distance classes are associated to the flow path within the entire catchment. By introducing these parallel basins we intend to merge similar values and therefore try to lower the average deviation within the distance class. This evaluation technique can be expressed as

30   follows: Let *B* be the number the sub-basins defined within an original watershed U. First step is to calculate the standard deviation σ (Eq. 2) within each sub-basin with SFL-classes based on flow length within the sub-basins.

In the second step the calculated σ-Values are transferred to the flow-length axis of the original watershed O. This is done for all sub-basins. Finally the new standard deviation σ(S) for the separated basin is calculated for each SFL-class as average of all σ values assigned to the same SFL-class:

$$\sigma(S)_{SFL} = \frac{1}{B} \cdot \sum_{j=1}^{B} \sigma_{j;SFL} \tag{5}$$

5   In comparison to the standard deviation of the unseparated basin σ(U) the success of the partition can be measured, e.g. as the quotient of these values.

**2.3.4 An algorithm for automated sub-basin ascertainment**

At this point two individual techniques, the objective function and the evaluation scheme have been established. Now these components have to be arranged in a reasonable order to perform an automated sub-basin ascertainment. Two basic

10   considerations have already been mentioned in the introduction of the tool:

1.  If low variance regions are present, these regions are clipped (by separation tool) from the rest of the basin.
2.  In case of present high variance regions solely subdivision at ramification and zonal classification are carried out in competition, since the origin of high variance is unknown

Obviously, before one of these cases can be rated true the objective function has to be called. This leads to the fundamental

15   sequence that first the objective function is called and based on the outcome, case 1 or 2, required tools are activated. Second case also requires the call of the evaluation scheme, since we have to decide which subdivision tool is more effective:

3.  Results of the technique (subdivision or zonal classification) yielding a lower σ(S) will be saved, other results will be discarded.

With this consideration only some sub-basins will comprise zonal classifications, giving an inconsistent result. In order to

20   obtain consistent results zones have to be calculated for all sub-basins:

4.  Zonal classification is called additionally in case only low variance regions are present. Thereby, independent from the need of σ reduction, all ascertained sub-basins will comprise a zonal classification.

Independent from all above mentioned considerations we assume that the presence multiple major streams within a single basin requires subdivision independent from variance compulsion. Major streams can be identified and subdivided by their

25   flow accumulation (see Appendix A for further detail). With this assumption we account for the need, e.g. to gauge or model these major streams separately. Giving the following consideration for the sequence:

5.  If multiple major streams are present, the subdivision at ramification is activated without calling the objective function.

Based on the outlined considerations the sequence of the ACS (Ascertainment by Characteristic Structure) algorithm has been

30   developed. It is depicted as a flow chart in Fig. 8.

Applied to a catchment with known coordinates of the outlet, the algorithm starts at this point. First it will check if multiple major streams are present and if necessary subdivide at ramification. Then the objective function is called and the required tool is activated. Each sub-basin will be analysed repeatedly until no SFL-class above $\Omega$ is left, or no further reduction can be generated. Please note that the described sequence is based on the presumption that shifts and breaks in the average of the characteristics along the SFL-axis are dispensable.

**3 Results**

Sub-basin ascertainment has been carried out on two characteristics: pore volume (total pore volume for Mulde, Regen and Main, AWC for Salzach (due to data availability, compare Fig. 2 & Fig. 3)) and surface slope. Please note that both applications and, hence, their results are disjoint.

Sub-basins and zones for all watersheds ascertained by the ACS-algorithm are depicted Fig. 9. The characteristic structure of $\sigma(\text{Unstructured})$ (blue lines) and $\sigma(\text{Structured})$ (red lines) as well as threshold $\Omega$ (according to Eq.(3)) for different weightings $e = 0$, $e = 0.5$ and $e = 1$ are shown in diagram. Drainage points of the defined sub-basins are classified by their derivation. Points at major ramifications are depicted as black triangles, while blue dots indicate points for separation of low variance regions and subdivisions at ramifications with order 1. These points are merged as category 1 points. Category 2 points indicate points of subdivision of higher order and hence indicate the separation of smaller contributing streams.

Although the success and differences between the outcomes of the algorithm are visible, we are still not able to quantify these observations. To make our results commensurable two numerical measures of success were developed. Both measures are based on the objective function and the evaluation scheme (Sec. 2.3.1 and 2.3.3) that imply minimisation of $\sigma$. In this cases (objective function and evaluation) only minor sets of distance classes are considered. To evaluate the performance of the algorithm within the entire catchment, all distance classes within the original, unseparated basin have to be considered.

Therefore Eq. (5) is applied to the stream-flow axis of the entire catchment and the ascertained sub-basins. The total reduction of the standard deviation $\sigma$ is e expressed as the subtraction of $\sigma(U)$ and $\sigma(S)$. To account for different expected values of the input data (in this case a lower value for slope than for storage) the reduction of $\sigma$ is normalized by the sum of initial standard deviation $\sigma(U)$. The first measure is therefore the relative reduction of variance within the entire basin $\alpha_1$:

$$\alpha_1 = \frac{\sum_{j=0}^{N_{sfl}} \sigma_j(U) - \sigma_j(S)}{\sum_{j=0}^{N_{sfl}} \sigma_j(U)} \tag{6}$$

With this measure we can quantify the success of a performed separation, by comparing remaining and initial heterogeneity. However, $\alpha_1$ does not necessarily tell if our target has been reached. For this purpose a second measure has been established as the ratio between the remaining standard deviations $\sigma(S)$ and $\sigma(U)$ above the threshold $\Omega$. For its calculation only distanceclasses comprising standard deviation greater than $\Omega$ are taken into account. The set of distance-classes fulfilling this requirement can be written as:

$$Z = [j_{sfl;1}, \ldots j_{sfl;n} \mid \sigma_j \leq \Omega] \tag{7}$$

Note that set Z is defined for the initial basin Z(U) and for the structured basin after application of ACS as Z(S). Second measure $\alpha_2$ can then be written as:

$$\alpha_2 = \frac{\sum\limits_{i \in Z(S)} \Omega - \sigma_i(S)}{\sum\limits_{j \in Z(U)} \Omega - \sigma_j(U)} \tag{8}$$

Values of proposed measures can be interpreted as follows: The higher the total reduction $\alpha_1$ and the lower the remaining variance above the threshold $\alpha_2$, the more heterogeneity could be compensated by the algorithm.

Keep in mind that these success rates examine the performance of the algorithm for a completed ascertainment. This is contrary to the evaluation of tools (Sect. 2.3.3) which only considers a single subdivision/separation.

Calculated $\alpha_1$ and $\alpha_2$ values for all applications are tabulated in Table 1. These values and the depiction of the results in Figure 9 shows that some applications were more successful than others, e.g. application on pore volume data in the Main catchment (nearly all classes below $\Omega$, best $\alpha_1$ and $\alpha_2$) compared to slope for the Mulde basin (lowest $\alpha_1$ and only few classes under $\Omega$). The density of close to stream zones is comparable for all basins, it ranges from 18 – 25 % for pore volume and 22 – 30 % for slope.

However, application on pore volume in all basins led to significantly higher success rates than for slope values. Another striking result is the outcome in the Salzach catchment. While the application of pore volume resulted had the lowest success, the application on slope has been more successful, especially $\alpha_2$ is with 67,2% significantly higher than for all other slope applications. The opposite is visible in the results of the Mulde, while application of pore volume led to average reduction for both parameters, slope application led to the lowest (total) reduction values. Generally it is visible (Fig. 9 and Tab. 1) that the standard deviation $\sigma(S)$ could be reduced in all basins and for both data applications.

**4 Discussion**

**4.1 Limiting factors for the algorithm**

In this section we will examine the results of spatial subdivisions obtained with the ACS algorithm. A special focus is set on catchment properties that affected the performance and hence define the outcome of the algorithm. Therefore we have to identify in which basins the algorithm performed well and in which it did not.

It is striking that applications on pore volume led to significantly higher reductions than applications on slope data. While the total reduction for pore volume ranges from 41 to 65 %, variance of slope could only be reduced in a range from 10 to 28 %. Though $\alpha_2$ ranges intersect, the average $\alpha_2$ for pore volume is superior to slope application. Nevertheless, low success rates

seem not to be limited to one catchment, but rather to the observed feature. These observations give first hints to the question which catchment property, or properties, affect the performance of the algorithm and hence define its results. Possible explanatory factors are the geomorphologic structure and/or the specific values of the considered features.

With our case study catchments we cannot prove these assumptions distinctly since structure and range of values are different. Especially the geomorphologic structures are nearly incomparable. To handle this problem we performed a resampling experiment. The basic concept is to examine structural identical catchments with a different range of featured values. First step is to examine the spatial organisation of pore volume in the Mulde as it is. Then we change the specific values of pore volume and repeat the analysis. If performance measures $\alpha_1$ and $\alpha_2$ are similar, the assumption about dependency of performance on the range of values has to be rejected.

In order to change featured values in a reasonable way, we did not change the values randomly but exchanged data of two natural catchments. Accounted by their similar size, the Mulde and the Salzach catchment were chosen for resampling. The exchange of pore volume between these two catchments has to retain the order and arrangement of the original catchments. Therefore we assigned high pore volume values of the original basins to high pore volume values in the exchange basin and applied this scheme to all values. Then the values were exchanged. The same exchange of values has been applied to the DEM, as the root of slope values. Figurate, the alpine structure of the Salzach has been shifted to soil and heights of the Ore Mountains, and the middle mountainous structure of the Mulde to an alpine soil and topography. If the performance in these resampled basins is identical to the performance in their origin basin (e.g. resampled AWC Mulde and original AWC Salzach) the assumption about dependency of performance on geomorphologic structure has to be rejected.

Resampled AWC values are shown in Fig.10. Stream network, SFL- and slope values were recalculated for the new DEM. Afterwards the ACS algorithm has been applied to the resampled basins. Results are shown in Fig. 11 and the success rates are given in Table 2.

If the characteristic structure for pore volume in the unseparated original catchments $\sigma(U)$ (Fig. 5 (Mulde) and Fig. 9 (Salzach) are compared to the resampled structure of $\sigma(S)$ (Fig. 11) a difference is visible. On the one hand the absolute values of standard deviation has changed and on the other hand the succession and arrangement of these values has changed. In case of slope data only the absolute amount of standard deviation has changed while the succession of values is nearly identical. A comparison of $\alpha_1$ and $\alpha_2$ values (Tab. 1 and Tab. 2) in the original and resampled basins shows that performance in the Mulde catchment slightly decreases or is stable, while it increased in the Salzach basin (with exception to $\alpha_1$ for pore volume which decreases by 8%).

As it can be seen in the characteristic structures, the range of $\sigma$ of the Mulde catchment is similar to the range of $\sigma$ in resampled Salzch basin and vice versa. Hence, the alteration of absolute standard deviations is directly caused by the performed exchange of values. Characteristic structures $\sigma(S)$ of resampled and original pore volume in the Salzach basin differ in the missing peaks of $\sigma(S)$. Generally resampled $\sigma(S)$ is more equally distributed than in the original basin. Contrary to this, $\sigma(S)$ in the resampled Mulde basin has a distinct peak of standard deviation at about 80-100 km distance from the outlet. These modification can be explained by the soil arrangement of the Salzach catchment.

Figure 12 shows a map of (original) AWC in the Salzach catchment in a larger scale. In this figure three regions within the catchment are marked with red squares. These regions comprise soil heterogeneities that can be described as small enclosures of high pore volume values. These enclosures differ significantly from their surrounding soil and do not follow the co-evolutional structure of the remaining basin. Since these enclosures cause variations they are visible in the characteristic

5    structure of the basin (right hand side of Fig. 12). With the transition of more uniform pore volume values from the Mulde basin, these enclosures vanish and hence the peaks in the structure. Consequently the high values of the enclosures are transferred to the Mulde, these spots are best visible in Fig. 10, located in the upper region on the right boundary of catchment. The placement of high value enclosures in the resampled Mulde catchment is the explanation for the reduction of performance $\alpha_2$. In the Salzach catchment, the total variance reduction decreased. Because the value of $\sigma(U)$ resampled and original are

10   similar, the lowered variance $\sigma(S)$ in the resampled basin causes this decrease. Nevertheless, the remaining variance above the threshold $\alpha_2$ is lowered by 13.1%. In the resampled Mulde basin $\alpha_1$ is nearly stable which is due to the fact that the enclosures are merged into a single, wider spot.

Following this discussion a first limitation can be stated: enclosure of values that do not follow the flow path arrangement of the catchment lower the ascertainment success rate. Moreover the difference between the actual values in the enclosure and

15   the surrounding values has an impact on the performance. The higher the difference the lower the success rate.

In case of the resampled slope values the success rates have changed only slightly (maximum alteration 8%) and the shape of characteristic structures for separated $\sigma(S)$ and unseparated basins $\sigma(U)$ are nearly identical in the original and resampled basin. Only the actual range of standard deviation has changed. This outcome indicates that the actual range of standard deviation does not affect the performance of the algorithm. Moreover, in accordance to the beforehand drawn conclusion the

20   geomorphologic structure can be identified as the main influencing variable defining outcome and performance of the algorithm. Hypothesis about dependencies between results and the range of featured values has to be rejected.

However, results also show that the performance for application on slope data is always inferior to applications on pore volume. The only reasonable explanation is that catchment characteristics directly bound to the structure (differences in heights define the flow path) cannot be encompassed properly by the algorithm in its present state.

25   In summary the application of the ACS algorithm to four different natural catchments and the resampled basins succeeded. Hypothesis about the influence of geomorphologic structure could not be rejected. Contrary to that we could show independence of the outcome from the range of catchment characteristic values. Nevertheless results indicate that the algorithm is not limited to a single catchment structure or range of values. As a major factor affecting the performance of the proposed algorithm the occurrence of soil enclosures, or in general term the occurrence of spots that differ significantly from the

30   surrounding values has been identified. Their affection on performance is dependent on the actual deviation between the enclosure and its surroundings.

**4.2 Spatial extend of zones**

Although the subdivision and separation tools are, in the sense of σ-reduction, mostly superior to zonal classification (what can be concluded from the high number of performed sub-divisions and, in relation to the initial basins, small sub-basins) a look at the spatial extend of zones is worthwhile. For this analysis we are looking at the "close to stream" zones, since "far from stream" zones are calculated as remaining cells. Results (Table 1 and Fig. 9) show that the average extent of "close to stream" zones is similar in all catchments and for both applications. Nevertheless, differences are visible between the sub-basins within these watersheds. Neighbouring and nested sub-basins reveal in some cases significant differences although their spatial allocation might suggest different. While some sub-basins are separated in small close to stream belts around the major streams and large "far from stream" zones, others incorporate extensive "close to stream" zones. The zone densities, calculated as the proportion of "close to stream" zones of the entire (sub-)basin, within all applications have an average of 24% but the range of densities moves from 0.5% to 67%. This observations leads to a fundamental question: Which properties of the basins surface, i.e. physiographic property regulates extend of these zones?

To answer this question we have to look into more detail of the results. Since the target of zonal classification is to reduce σ of the sub-basin by separating into two sets of values, there are two ways to cope with this task. Either the "close to stream" zone tries to cover as much cells as possible with a minimum of variance, or it tries to embrace regions of the sub-basin casing a maximum of variance with a minimum of coverage. These two operations of zonal classification can be called "Minimisation" and "Maximisation". In Fig. 13 the usage of these operations is shown on the example of the Mulde. Extend of zones is perceivable not bound to the operation. If we now take the relationship between standard deviations in "close to –" and "far from stream" into account, see right of Fig. 13, a pattern becomes visible. While narrow zone belts around streams tend to comprise a high quotient of standard deviations, extensive zone are mostly present if the quotient is low. Results of the other basins are not shown here, but imply identical interpretations.

These findings can answer the initial question: Extend of zones is bound to the presence of patterns around the major streams within a catchment. In the absence of a clear stream orientated pattern, "close to stream" zone get more extensive.

**4.3 Comparison to gauging networks**

After the evaluation of functionality the usefulness of the proposed algorithm has to be evaluated. One possible field of application is the use for semi-distributed hydrological models or as reference subdivision for the examination of existing gauging networks. In both applications defined sub-basins, defined by modelling nodes or gauges, are defined to reduce the catchments heterogeneity and to model or observe discharge from mostly homogenous regions. Obviously, existing gauging networks are a result of multiple considerations and requirements and in some parts of the basin tend to be denser than required to catch the natural heterogeneity of a single catchment characteristic. On contrary, due to its multiple requirements, the gauging network is not orientated on a single characteristic and will always represent a compromise of possible decisions. A comparison to results of the ACS algorithm is therefore not in all cases fair. Nevertheless, differences in the number of

separation points, or the reduction of variance can give hinds to decision makers for possible new gauging positions. The usefulness for the decision maker is bound the informational value of the specific catchment characteristic for runoff generation processes.

However, for our purpose, the evaluation of the ACS algorithm we assume that pore volume and slope are crucial information.

5 To evaluate the power of the ACS algorithm two comparisons are performed. First, a subdivision based on the gauging network will be compared to the obtained basins (without ACS zones) introduced in Sec. 3. With this analysis it will be possible to evaluate the power of defined separation points. A Second comparison will use the basins and zones introduced in Sec. 3 compared to a subdivision based on the gauging network and a zonal partition by land cover. Based on the suggestions by Lindström et al. (1997) land cover has been merged into two categories: field and forest. Additionally a third zone Rock / Bare

10 soil has been introduced, which has been considered necessary for the Salzach catchment. Although the simple land cover omits the exact allocation of the zones, due to its simplicity and common usage it appeared to be adequate. By introduction of this classification a fair benchmark has been set up for comparison. The gauging network and defined land cover zones are shown on the left of Fig. 14

The measure of success (Eq. 5) has been applied to these subdivisions and the characteristic structure of $\sigma(S)$ for pore volume

15 and slope were determined. Characteristic structures for gauging network (blue) and land cover (green) partition are depicted in the middle and on the right of Fig. 14. For visual comparison the characteristic structure of ACS basins (red) are added in the figure. Success rates $\alpha 1$ (Eq. 6) and $\alpha 2$ (Eq. 8) were calculated and presented in Tab. 3.

If we compare results for ACS-basins only and the gauging network in Tab. 3 it is visible that the success rates are significantly higher for ACS basins. In average the reduction of the standard deviation $\alpha_1$ for both variables is 50% higher and the remaining

20 standard deviation above the threshold $\alpha_2$ is 32% lower. This difference has two reasons: On the one hand for some basins more separation points than gauging points have been defined. On the other hand the adjustment of separation points on the specific variable itself led to less points than the gauging network. The first case is a hind for a necessary extension of the gauging network (or node-network) and the second case either approves the existing network or indicates better positions for existing gauges / nodes.

25 For a second analysis the analysed sub-basins are subdivided into zones by ACS classification and land cover zones. Performance of sub-basins ascertained by gauging network and land cover zones is treated as benchmark for the ACS algorithm. In Fig. 14 it is visible that ACS (red) results give lower or equivalent $\sigma(S)$ values to the benchmark partition (green) for pore volume application. This observation is confirmed by success rates given in Tab. 1 and 3. Average advantage of the ACS is 42% for $\alpha_1$ and 66% for $\alpha_2$. Contrary to this result, application on slope shows that the benchmark is, with exception

30 of the Regen catchment, superior to the ACS results (15% for $\alpha_1$ and 30% for $a_2$).

Since the comparison of sub-basins solely showed that ACS is superior to the benchmark, the explanation for this result is the usage of land-cover zones. These zones are independent from the structure of the basins and could possibly give a better, or at least an alternative way to assess network bound parameters. This results indicates a need for further improvement of ACS-classification. For example a third category based on land use or heights might be beneficial.

Ascertained sub-basins have been compared with the elaborate technique of stream flow orientated standard deviation, introduced by ourselves. Obviously this is not an acceptable validation of the proposed method. Since a fully-fledged validation is dependent on the purpose, e.g. performance in a hydrological model, regionalisation scheme etc. (which would go beyond the scope of this paper) we have to address the validation on a basic level. But how are different spatial resolutions commonly compared? A possible and plain answer is to compare the average of variation (for considered characteristics) within the sub-basins.

For this analysis we took the sub-basins ascertained by ACS and the gauging network and omitted the zones. For each sub-basin the plain standard deviation of pore volume and slope has been calculated. Table 4 shows the area weighted averages of these values per catchment and the relative advantage of ACS sub-basins. The advantage ranges from 1.4 to 15.1% which is a rather low advantage. Yet it has to be taken into account that zones have been neglected and the attention of this study is set on the flow path.

**4.4 Definition of physiographical types**

As mentioned in the introduction the proposed algorithm accounts for the spatial organisation of the catchment and the interaction with the catchment characteristics. Hence, its results should incorporate information about the spatial organisation and information about the lateral flow path. Grayson and Blöschl (2001) pointed out that these information could give insight to processes of lateral flow distribution. Therefore it seems worthwhile to establish a similarity measure for sub-basins ascertained with the ACS-algorithm. In this section we will give a prospect on a possible classification scheme which could be used for regionalisation studies.

The test of general applicability (Sec. 3) yielded sub-basins and zones for different catchment characteristics. To assess the similarity of these results (to assess the spatial organisation) these results have to be combined. The simplest way to combine the outcome of different input data, and retain the univariate objective function, is to run the algorithm in turn over all input data and use ascertained sub-basins as pre-subdivision. Afterwards a combined set of sub-basins is obtained that accounts for all input data. In order to obtain zones for these sub-basins, the algorithm has to be rerun for all input data, using the beforehand obtained sub-basins. Individual zone networks are available for each catchment feature after this procedure.

To combine them it seems worthwhile to overlay the obtained data sets. Figure 15 shows the overlay of zones for slope (yellow) and pore volume (blue) for all catchments used in this study. Intersecting zones appear greenish. It is now up to the user to decide how these zones are combined. Either they can be combined through logical operators (intersection or union) or as weighted averages by means of weights of the input data or by the reduction of variance of each zone.

Beside the combination of zones they could be used for the assessment of spatial organisation of catchments. After the application of the algorithm we assume that each sub-basin certainly represents a homogeneous natural unit. Defined zones indicate the structure of the input data with respect to drainage system within this unit (compare Sec. 4.2). Following the definition of Winter (2001) a hydrological landscape is determined by the interaction of a physiographic feature and the

hydrologic system. The physiographic part is defined by its land surface form, the geological system and its climate. The hydrologic system defined by surface, ground and atmospheric water.

With the conducted case studies, we are able to use the slope data as information about the flow energy and the pore volumes as information about the pedologic system and, hence, are able to identify similar patterns of surface morphology. Similar patterns are classified into physiographical types.

If Fig. 15 (left side) is examined with this perspective four types of zonal intersections are visible:

- Type A: Narrow slope and pore belts. Soil is arranged in valleys along main rivers. Especially visible in map excerpt I in Fig. 15 (right side).

- Type B: Soil belts and extensive slope zones. Soil patterns are bound the drainage system, with disseminated side-arm valleys, or plains.

- Type C: Slope belts and extensive soil zones. Valleys around the main rivers with similar to hillslope soils.

- Type D: Plains or disseminated side-arm valleys with no predominant soil pattern. Shown in map excerpt II in Fig. 15 (right side).

In map excerpt A in Fig. 15 we can see a wide valley, with sloping hillsides, the AWC follows this patterns with higher values in the stream valley. "Close to stream" zone densities (left side of Fig. 15) are low and shaped like a single line. In contrast, map excerpt B shows a shallower, disseminated valley with no soil pattern. These two excerpts are good examples for Type A and D sub-basins.

To derive a physiographic type for a certain sub-basin its zone densities have to be rated has high or low. Therefore a k-Means analysis has been carried out for an, as much as possible, impartial differentiation between high and low densities. Boundary values were found in a range from 22% to 27% zone density. A scatterplot of zone densities in the Salzach catchment and the calculated k-means boundaries are shown in Fig. 16. The four sectors can be interpreted as the four defined physiographical types: Type A in the lower left, Type B upper left, Type C lower right and Type D in the upper right.

This classification schemes has been applied to all catchments. The obtained maps of present physiographical types are shown in Figure 17. In the Mulde catchment types A and B are predominant while in the Regen catchment type B is dominant solely. While the Main catchment is dominated by C and D types, the Salzach catchment is split into a Type A dominated part and mixed section.

In summary the automatically ascertained sub-basins and zones have been used to categorize regions of catchments into different physiographical types. These types were designed to represent different surface and soil patterns. For the actual categorisation we used the density of defined zones and used gathered information (Sec. 4.2) about the link of stream network patterns and zone density to derive a classification scheme. However, the absence of an impartial threshold required the (more or less) subjective choice a threshold value for classification of the sub-basins. Therefore the presented results have to be considered as a prospect to future work and possible applications of the algorithm.

**5 Conclusions**

A new approach for the analysis of watersheds based on the interaction of catchment characteristics and the flow path has been introduced in this study. Based on this method an algorithm has been proposed for automated and impartial sub-basin ascertainment. This new methods and algorithm detect and incorporate the spatial organisation of the watershed into the definition of sub-basins and zones. General Applicability on different catchment characteristics and different catchment structures has been shown in a case study. The proposed ACS algorithm has been applied to topography and soil data of four meso-scale watersheds. Results showed that the performed subdivisions delivered a reasonable number of sub-basins and could significantly reduce the heterogeneity of the considered characteristic within these sub-basins. A detailed analysis, incorporating a numerical resampling experiment, could additionally detect some limitations for the ACS-algorithm. Especially the occurrence of soil enclosures, incorporating properties that differ significantly from their surrounding soil, could be identified as interfering. But the occurrence of these enclosures does not always led to significant reduction of the success rate, but rather the heights of the difference between the enclosure and its surrounding were identified to be important. Performed applications and experiments could show that geomorphology has an impact on the performance of the algorithm. But its general applicability is not limited to a single catchment structure. Additionally we could show that characteristics directly bound to the catchment structure (i.e. flow path) could not be encompassed sufficiently. Although limitations have been identified, the proposed techniques performed well in our case studies.

Its performance, in sense of flow-path orientated standard deviation has been compared to the common strategy of sub-basin definition based on the gauging network and land cover classification. It could be shown that sub-basins ascertained by the proposed algorithm inhibit a significantly lower variation along the flow path. Furthermore a prospect on the usage of the algorithm and its outcomes for modelling and regionalisation purposes has been given. In comparison to the employed benchmark methods (and other methods) some advancements of the proposed method and algorithm can be pointed out. First, the underlying method of characteristic structures which accounts for the occurrence of catchment characteristics by their distance to the outlet differentiated between stream flow length and over land flow length. This approach employs more information about a basin than other common pattern identification schemes, like point-to-point comparisons or optimal logical alignment (Grayson and Blöschl, 2001). Nevertheless, the full benefits of these advancements for different hydrological purposes is still outstanding and will be addressed in upcoming research.

Second, the ACS-algorithm incorporates three techniques for sub-basin ascertainment and accounts for the interaction and succession of different catchment characteristics. This is a major advancement to common subdivision methods which are based only on the stream network, via Pfafstetter codification (Verdin and Verdin, 1999) or by Strahler order (Rodríguez-Iturbe and Valdés, 1979). Other methods that do account for surface characteristics are usually HRU concepts (Winter, 2001; Wagener et al., 2007; Schumann et al., 2000; Winter, 2001) that neglect the allocation of coherently alike regions and their succession within the catchment. Although ACS incorporates a zonal classification that is comparable to an HRU concept, the

defined zones do retain the order within the sub-basin in sense of the flow path. The concept is most likely comparable to the concept of Nobre et al. (2011).

More specifically, the ACS-algorithm defines two zone types within each sub-basin to account for stream network orientated patterns. Since we assumed that the algorithm produces homogeneous natural units the density of zones has been used to define

5    independent physiographical types as part of hydrological landscapes (Winter, 2001). As a measure of physiographical similarity we used the spatial extend of "close to stream" zones, since they indicated the presence of stream network orientated patterns. In this paper only two variables were used and hence only two zone types and four physiographical types were identified. A additional variables are the height above the nearest drainage (Nobre et al., 2011) in order to account for plateaus within the catchment and the hydraulic conductivity of soils.

10   However, the obtained results only give physiographical similarity of the sub-basins. Future work has to account for the hydrologic system and has to compare the hydrologic similarity to the physiographic similarity. With this intersection the utility of the proposed method for the description hydrological landscapes could be evaluated and elaborated to a catchment classification scheme. In contrast to other schemes (e. g. 2010; Sawicz et al. 2011 and Winter, 2001) the derived landscape types are not derived by statistics about the sub-basin but by its spatial organisation.

15   **Appendix A**

In this Appendix details to the proposed ACS-tools are given for further understanding of the algorithm. Each tool will be addressed separately.

**Separation tool for low variance regions**

[revised manuscript text omitted]

Because we do not know the dominant spatial pattern of the input data the search for an optimal extent of the "close to stream" zone is done iteratively. The iteration employs two variables: the reduction of the Strahler order $s_R$ from the maximum occurring Strahler order $M_S$ and the width of the zones Δo, expressed as multiple of cell width. Parameter ranges are [0; $M_S$-1] for $s_R$ and [0, 5] for Δo, respectively. Cells draining into streams cells with Strahler order equal or lower than $M_S$-$s_R$ and an OFL value equal or lower than Δo·Δx are marked as "close to stream". In Fig. A3 a sequence of the iteration is shown for the entire synthetic catchment. After each iteration, the standard deviation σ (Eq. (2) & (5)) is calculated for each zone and subsequently averaged. The parameter combination ($s_R$, Δo) with the lowest averaged standard deviation is chosen for final classification. In Figure 7, an example for a chosen zonal classification for the synthetic catchment is shown. The ACS will define results of the technique leading to the highest reduction of the standard deviation σ as superior and omits the inferior result.

Please note that the algorithm has the possibility to neglect the usage of zonal classification. If the calculated averaged σ of the zones is equal to or higher than σ of the unseparated data, the whole basin will be marked as zone type "none".

**Table 1: Results of applications of *ACS*. Number of ascertained sub-basins, normalized reduction of standard deviation $\sigma$ and density of close to stream zones.**

| Catchment | Pore Volume | | | | Slope | | | |
| --- | --- | --- | --- | --- | --- | --- | --- | --- |
| | No. of Basins[-] | $\alpha_1$ (Eq. 7) [%] | $\alpha_2$ (Eq. 9) [%] | Close to stream zones [%] | No. of Basins [-] | $\alpha_1$ (Eq. 7) [%] | $\alpha_2$ (Eq. 9) [%] | Close to stream zones [%] |
| Mulde | 70 | 52.2 | 69.8 | 22.3 | 80 | 10.6 | 49.0 | 22,1 |
| Main | 59 | 64.5 | 82.9 | 25.8 | 22 | 19.3 | 26.9 | 26.5 |
| Regen | 17 | 52.8 | 70.2 | 18.8 | 24 | 16.1 | 28.9 | 30.1 |
| Salzach | 24 | 41.0 | 56.8 | 20.9 | 38 | 18.4 | 67.2 | 24.4 |

**Table 2: Normalized reduction of standard deviation $\sigma$ for resampled basins**

| Catchment | Pore Volume | | Slope | |
|---|---|---|---|---|
| | $\alpha_1$ [%] | $\alpha_2$ [%] | $\alpha_1$ [%] | $\alpha_2$ [%] |
| Mulde (res) | 54.7 | 69.2 | 8.5 | 51.1 |
| Salzach (res) | 33.4 | 32.8 | 20.9 | 68.0 |

**Table 3: Normalized reduction of standard deviation $\sigma$ for sub-basins based on gauging network, ACS-basins and gauges and land cover**

| Catchment | Pore volume | | Slope | | No. of Gauges |
|---|---|---|---|---|---|
| | $\alpha_1$ [%] | $\alpha_2$ [%] | $\alpha_1$ [%] | $\alpha_2$ [%] | |
| *Gauging network* | | | | | |
| Mulde | 24.2 | 57.7 | 9.4 | 43.3 | 40 |
| Main | 41.9 | 81.1 | 14.0 | 15.8 | 46 |
| Regen | 18.9 | 54.9 | 6.5 | 17.4 | 20 |
| Salzach | 30.3 | 49.6 | 9.6 | 58.6 | 33 |
| *ACS-basins only* | | | | | |
| Mulde | 39.2 | 59.4 | 13.7 | 44.9 | 70/80 |
| Main | 55.8 | 79.8 | 14.7 | 19.8 | 59/22 |
| Regen | 41.3 | 61.4 | 11.5 | 27.2 | 17/24 |
| Salzach | 31.3 | 48.1 | 11.9 | 61.3 | 24/38 |
| *Gauging network & land cover* | | | | | **Occ. zones** |
| Mulde | 35.3 | 62.8 | 14.5 | 43.1 | 2 |
| Main | 48.9 | 84.6 | 19.8 | 18.8 | 2 |
| Regen | 25.1 | 58.5 | 14.3 | 19.5 | 2 |
| Salzach | 38.4 | 64.6 | 21.6 | 66.8 | 3 |

**Table 4: Averages of plain standard deviation for sub-basins ascertained by ACS and gauges**

| Catchment | Pore Volume [mm] | | | Slope [°] | | |
|---|---|---|---|---|---|---|
| | ACS | Gauges | Diff. | ACS | Gauges | Diff. |
| Mulde | 131.7 | 142.7 | 7.7% | 2.7 | 3.2 | 15.1% |
| Main | 155.5 | 158.8 | 2.1% | 3.9 | 4.1 | 1.4% |
| Regen | 90.5 | 105.7 | 14.4% | 4.2 | 4.1 | 3.0% |
| Salzach | 155.5 | 158.9 | 2.1% | 3.9 | 4.1 | 1.4% |

[Figure]

**Figure 1: Digital Elevation models of the Regen (upper left) and upper Main (upper right) both in Bavaria, Germany, Salzach (lower left) in Salzburg, Austria, and the Mulde (lower right) in Saxony, Germany.**

[Figure]

**Figure 2: Values of total pore volume in the catchment of Main (left), Regen (centre) and available water capacity for catchment of Salzach (right)**

[Figure]

**Figure 3: Available water capacity in the Mulde catchment (left) and calculated characteristic structure (right)**

[Figure]

5    **Figure 4: Average AWC in OFL classes in the Mulde catchment**

[Figure]

**Figure 5: Standard deviation σ(AWC) in SFL-classes and Threshold values $\tau_S$ for different values of $e$, in the Mulde catchment**

[Figure]

**Figure 6: Flow direction and Strahler order (left), SFL-data and –classes (middle) and example input data (right)**

[Figure]

**Figure 7: Answers of the objective function (left) and separation (upper right) and subdivision techniques (lower right) in the synthetic catchment**

[Figure]

**Figure 8: Sequence of the ACS-algorithm**

[Figure]

**Figure 9: Results of application of *ACS* for catchments of the Mulde, Main, Regen and Salzach (from top to bottom), sub-basins based on pore volume (left) and slope (right). Comparison of σ(U) and σ(S) for each application (red and blue lines).**

[Figure]

**Figure 10: Resampled AWC values for Mulde and Salzach catchment (scale = 1:2500000)**

**Figure 11: Results of application of the algorithm for resampled catchments of the Mulde and Salzach (from top to bottom), sub-basins based on resampled pore volume (left) and slope (right). Comparison of σ(U) and σ(S) for each application (red and blue lines).**

[Figure]

**Figure 12: AWC of the Salzach catchment and the characteristic structure of σ (U) and σ (S). Red marked and numbered areas incorporating high value enclosures**

[Figure]

**Figure 13: Operation of zonal classification (left) and proportion of σ of "close to –" and "far from stream" zones (right) in the Mulde catchment**

[Figure]

**Figure 14: Subdivisions based on gauging network & zonal classification and characteristic structures of σ(pore volume) and σ(slope) (left to right) for catchments of the Mulde, Main, Regen and Salzach (top to bottom)**

Excerpt I    Excerpt II

Basins Slope [°]
Max.: 50.0
Min.: 0.0

Total pore volume [mm]
Max.: 938
Min.: 184

Zones

Slope zones

Pore volume zones

Kilometers
0  10  20    40    60    80

**Figure 15 Left: Overlay of calculated zones for pore volume (blue) and slope (yellow), intersecting zones are shown in green. Right: Map excerpts of slope and pore volume data for indicated areas I and II**

[Figure]

5    **Figure 16: Scatterplot of zone densities and results of k-means cluster analysis for the Salzach catchment and assigned physiographic types (A-D)**

[Figure]

[Figure]

**Figure 17: Identified physiographic types in case study catchments**

[Figure]

**Figure A2: Characteristic structure of Flow Accumulation in the catchment of the Mulde**

[Figure]

**Figure A3: Sequence of parameter iteration for entire synthetic catchment. Iteration from upper left to lower right**